# New Constructed EEM Spectra Combined with N-PLS Analysis Approach as an Effective Way to Determine Multiple Target Compounds in Complex Samples

**DOI:** 10.3390/molecules27238378

**Published:** 2022-12-01

**Authors:** Zeying Li, Na Feng, Xinkang Li, Yuan Lin, Xiangzhi Zhang, Baoqiong Li

**Affiliations:** School of Biotechnology and Health Sciences, Wuyi University, Jiangmen 529020, China

**Keywords:** new constructed spectra, N-way partial least squares, partial least squares, quantitative analysis, complex samples

## Abstract

Excitation–emission matrix (EEM) fluorescence spectroscopy has been applied to many fields. In this study, a simple method was proposed to obtain the new constructed three-dimensional (3D) EEM spectra based on the original EEM spectra. Then, the application of the N-PLS method to the new constructed 3D EEM spectra was proposed to quantify target compounds in two complex data sets. The quantitative models were established on external sample sets and validated using statistical parameters. For validation purposes, the obtained results were compared with those obtained by applying the N-PLS method to the original EEM spectra and applying the PLS method to the extracted maximum spectra in the concatenated mode. The comparison of the results demonstrated that, given the advantages of less useless information and a high calculating speed of the new constructed 3D EEM spectra, N-PLS on the new constructed 3D EEM spectra obtained better quantitative analysis results with a correlation coefficient of prediction above 0.9906 and recovery values in the range of 85.6–95.6%. Therefore, one can conclude that the N-PLS method combined with the new constructed 3D EEM spectra is expected to be broadened as an alternative strategy for the simultaneous determination of multiple target compounds.

## 1. Introduction

Fluorescence has the advantages of being non-destructive, simple, fast, having higher selectivity and sensitivity over extinction spectroscopic techniques, and having an inherently multidimensional character [1]. Excitation–emission matrix (EEM) fluorescence spectroscopy can provide complete fluorescence information on measured samples by covering a wide range of different excitation and emission wavelengths. In recent years, the EEM technique has been gaining widespread analytical acceptance in many fields such as environmental [2], cell culture media [3], cosmetics [4], food [5,6], and so on. For a given sample, a two-dimensional signal that contains peaks from all of the excited fluorophores can be obtained [7]. For a set of samples, complex three-way data structures with the dimensions of emission wavelength × excitation wavelength × samples can be formed. On the one hand, such three-way data structures may provide new opportunities for extracting useful information for further analysis; on the other hand, they also bring a challenge for data analysis.

An effective way of extracting characteristic information is to develop useful chemometrics or properly employ the proposed methods. In the past years, a variety of chemometric methods have been used to analyze EEMs, such as parallel factor analysis (PARAFAC) [8], alternating trilinear decomposition (ATLD) [9,10] and its variants, N-way partial least squares (N-PLS) [11], multivariate curve resolution-alternating least squares (MCR-ALS) [12], and unfolded partial leasts quares/residual bilinearization (U-PLS/RBL) [13]. The attractive merit of the “second-order advantage” of these methods makes sure that multiple target compounds can be accurately quantified even in the presence of unknown interferences [14,15]. Moreover, image moments such as the Zernike moment [16], Wavelet moment [17], Krawtchouk moment [18], and Tchebichef moment [19] have also been successfully applied in EEM spectra analysis. Owing to the multiple advantages of the image moments, such as inherent invariance property, image description capability, and multi-resolution capability, the overlapping, scatterings, and other interference signals did not affect the analytical results. Therefore, these advanced chemometric methods open a window into exploring complex mixtures and avoid the traditional costly, complex, and time-consuming analytical techniques.

As for EEM spectral analysis, previous studies usually employed the ranges of the analytes to construct calibration models. Compared with the models on the EEM spectra, modeling with the characteristic spectra selected via some effective methods might give models with better performance. More efforts are needed to extract effective and characteristic spectra for establishing quantitative models and predicting the content of the target analytes in complex samples.

This study aims to extract characteristic spectra from the original EEM spectra to form the new constructed 3D EEM spectra and to explore the potential of the N-PLS method on the new constructed 3D EEM spectra to quantify multiple target compounds. To our knowledge, there have been no articles that report on the above strategy for quantitative analysis purposes. Moreover, comparisons were made between the results obtained from applying the N-PLS method to the original EEM spectra and the PLS method to the extracted maximum spectra in the concatenated mode.

## 2. Data Sets

### 2.1. Data Set 1

This data set was obtained from a public data set (http://www.models.life.ku.dk/joda/prototype, accessed on 2 April 2021). The samples consisted of three chemicals (valine–tyrosine–valine (Val–Tyr–Val), tryptophan–glycine (Trp–Gly), and phenylalanine (Phe)) which were measured via fluorescence spectroscopy in the form of EEM [20]; the corresponding data set was stored in a three-way array with the sizes of 251 (emission wavelength points) × 21 (excitation wavelength points) × 26 (sample number). Figure 1 shows the EEM spectra of the three target compounds. As can be seen from this figure, the three compounds overlap.

### 2.2. Data Set 2

The EEMs of 25 samples were measured with a Horiba FluoroMax-4 spectrofluorometer. For the 25 samples, the first 23 samples were the standard samples only containing magnolol and honokiol, the 24th sample was an extract of Magnoliae Cortex (a kind of traditional Chinese medicine), and the 25th sample was a spiked sample whichwas spiked with suitable amounts of standard magnolol and honokiol. A 1.00 cm quartz cell was used. The excitation wavelengths ranged from 234 nm to 370 nm with 2 nm increments, and emission wavelengths ranged from 300 nm to 500 nm increments. The corresponding data set was stored in a three-way array with sizes of 101 (emission wavelength points) × 69 (excitation wavelength points) × 25 (sample number). The EEM spectra for the two target compounds (Figure 2A,B) and the Magnoliae Cortex (Figure 2C) are illustrated. As can be seen from this figure, the spectra of magnolol and honokiol overlapped. There are some established ways to deal with scatterings in the EEM spectra: in the present study, we used an interpolation method to solve this problem. In more detail, the scatterings are first removed and then the missing data are filled through interpolation using new data consistent with the rest of the EEM spectra [21]. The spectrum without scatterings is illustrated in Figure 2B.

## 3. Data Analysis

### 3.1. The New Constructed 3D EEM Spectra

The steps to obtain the new constructed 3D EEM spectra are illustrated subsequently. Firstly, the maximum excitation and emission spectra for each target compound were determined individually based on the EEM spectrum of the corresponding pure solution (Figure 3A). The data of each extracted spectrum are colored in Figure 3C. Then, the extracted maximum excitation spectrum and maximum emission spectrum were arranged as a constructed spectrum (Figure 3B) for a target compound, and its representation of the data matrix is illustrated in Figure 3D.

As for the EEM spectra that consisted of multiple target compounds, the maximum excitation spectrum and maximum emission spectrum were extracted (as illustrated in the top of Figure 4) for each target compound, respectively, in which the maximum excitation spectrum and maximum emission spectrum were determined based on the EEM spectrum of the corresponding pure solution (Figure 3A). Then, the extracted maximum excitation spectrum and maximum emission spectrum were combined for each target compound (as illustrated in the middle part of Figure 4). Finally, the combined spectra of each target compound were constructed again into three-dimensional (3D) spectra (as illustrated in the bottom of Figure 4). We named the constructed 3D spectra as the new constructed 3D EEM spectra. The new data representation of a sample (D_C1_) can be constructed as illustrated in the top of Figure 5, and the new data representation of all samples (D_C_) was constructed as illustrated at the bottom of Figure 5.

In the present study, for Data Set 1, the new EEM spectra of 26 samples were arranged in a matrix with the sizes of 3 (the number of target compounds) × 272 (the sum of excitation wavelength points and emission wavelength points) × 26 (the number of samples). For Data Set 2, the new EEM spectra of 25 samples were also arranged in a matrix with the sizes of 2 (the number of target compounds) × 170 (the sum of excitation wavelength points and emission wavelength points) × 25 (the number of samples). For each data set, the matrix (D_C_ in Figure 5) was employed as the input data of the N-PLS method. In our opinion, in this way, not only the useful and characteristic information can be extracted, but also the interference information can be eliminated to some extent. It is important to note that the partial maximum excitation and emission spectra of each target compound can be extracted as long as the signals are not completely overlapping. Therefore, satisfactory analytical results can be obtained with the application of some advanced chemometric methods based on the new constructed 3D EEM spectra.

### 3.2. N-PLS Method

The N-PLS method is an extension of PLS to handle multi-way data [22] that can decompose the data array (dependent variable) to parallel factor analysis and then predict independent variables such as concentrations [23]. Bro et al. described the algorithm in detail in the original literature [24]. The N-PLS method has successfully been used for multi-way data modeling in many areas, such as environmental [25], food science [26], and so on. The determination of optimum latent variables (*LV*s) is an important step for the establishment of a stable N-PLS model with high accuracy. In the present study, in order to avoid overfitting or underfitting problems and to improve the predictive performance of the established models, leave-one-out cross-validation was used to select the optimum number of *LV*s. The N-PLS method was implemented using the MATLAB toolboxes (http://www.models.life.ku.dk/algorithms, accessed on 5 June 2021).

### 3.3. PLS Method

PLS is a multivariate statistical analysis method which aims to find out the relationship between prediction matrices and response matrices [27]. The PLS method has been successfully employed in the determination of multiple components in food [28], drugs [29], and many other fields. Like the N-PLS method, the selection of the largest potential variables (*LV*s) is particularly important [30]. Cross-validation is often used to optimize the number of *LV*s to ensure the predictive ability of the model [31]. In the present study, leave-one-out cross-validation was used to select the optimum number of *LV*s.

### 3.4. Regression Modeling and Evaluation

As for the two employed data sets, the calibration and test sets were divided. The models were developed based on the calibration set and then used to predict the test set. The statistical parameters, including the correlation coefficient of calibration (*R_c_*), leave-one-out cross-validation (*R_loo-cv_*), and prediction (*R_p_*) as well as the root mean squared errors of calibration (*RMSEC*), leave-one-out cross-validation (*RMSECV*), and prediction (*RMSEP*), were calculated to estimate the reliability and accuracy of the established models and their predictive ability in practical applications. To be specific, *RMSEC* explains how good the model is, *RMSECV* explains the ruggedness, and *RMSEP* illustrates the predictive ability of the model [32]. An excellent model has good model precision and prediction ability with a higher *R_p_* and a lower *RMSEP* [33]. Furthermore, to test the reliability of the calibration results visually, the calculation concentrations and the experimental concentrations of the target compounds were plotted. Once a calibration model was established, the *RMSEC* was defined as follows:(1)$$RMSEC=\sqrt{\frac{\sum_{i=1}^{n}{\left(y_{i}-{\tilde{y}}_{i}\right)}^{2}}{n}}$$
where $y_{i}$ is the experimental value of the *i*th sample, ${\tilde{y}}_{i}$ is the calculated value of the *i*th sample which is predicted by the model directly, and *n* is the number of samples in the calibration set. *RMSECV* can be calculated as follows:(2)$$RMSECV=\sqrt{\frac{\sum_{i=1}^{n}{\left(y_{i}-{\hat{y}}_{i}\right)}^{2}}{n}}$$
where $y_{i}$ is the experimental value of the *i*th sample, ${\hat{y}}_{i}$ is the predicted value of the *i*th sample which is estimated by the model when the *i*th sample is removed (the process is repeated for every sample in the training set), and *n* is the number of samples in the training set. *RMSEP* can be calculated as follows:(3)$$RMSEP=\sqrt{\frac{\sum_{i=1}^{m}{\left(y_{j}-{\tilde{y}}_{j}\right)}^{2}}{m}}$$
where $y_{j}$ is the experimental value of the *j*th sample and ${\tilde{y}}_{j}$ is the estimated value of the *j*th sample in the test set, whereas *m* is the number of samples in the test set.

## 4. Results and Discussion

### 4.1. N-PLS Method on New Constructed 3D EEM Spectra

#### 4.1.1. The Results of Data Set 1

The whole data set was divided into a calibration set (19 samples) and a test set (7 samples). The optimum number of *LV*s for each N-PLS model was determined according to the corresponding minimum *RMSECV* value. Here, the optimum number of *LV*s for the determination of Val–Tyr–Val, Trp–Gly, and Phe can be selected as three, five, and five, respectively. Then the calibration models for the three target compounds can be established.

After the calibration step, the established models were used to predict the concentrations of Val–Tyr–Val, Trp–Gly, and Phe in the test set. The statistical parameters for the evaluation of the established models and their predictive abilities are summarized in Table 1. For the convenience of observation, the calculation concentrations are compared with experimental concentrations in Figure 6. As illustrated in Table 1 and Figure 6, good results were obtained. The established models have high values of *R_c_* and *R_loo-cv_* (higher than 0.9872) and low values of *RMSEC* and *RMSECV* (lower than 0.2335), showing that the established models have good linearity, high reliability, and accuracy. Moreover, the values of *R_p_* and *RMSEP* are higher than 0.9983 and lower than 0.1111, respectively, indicating that the established models have a good predictive ability.

#### 4.1.2. The Results of Data Set 2

As for this data set, the whole data set was divided into two groups randomly, the calibration set includes 17 samples and the test set includes 6 samples. The optimum number of *LV*s for the N-PLS model was determined similarly to Data Set 1. The optimum number of *LV*s for the determination of magnolol and honokiol were determined as four and four.

The statistical parameters for the evaluation of the established models and their predictive abilities are summarized in Table 2. For the convenience of observation, the calculation concentrations are compared with the experimental concentrations in Figure 7. As illustrated in Table 2 and Figure 7, good results were obtained. The established models have high values of *R_c_* and *R_loo-cv_* (higher than 0.9780), and low values of *RMSEC* and *RMSECV* (lower than 4.7487), showing that these models have good linearity, high reliability, and accuracy. Moreover, the values of *R_p_* and *RMSEP* are higher than 0.9906 and lower than 4.5444, respectively, indicating that the established models have a good predictive ability.

### 4.2. Comparison with Other Methods

In order to further validate the performance of the proposed strategy, the N-PLS method has been applied to the original EEM spectra, and the PLS method has been applied to the extracted maximum excitation and emission spectra in the concatenated mode, in other words, putting together extracted maximum excitation and emission spectra for all target compounds in the single row vector).

#### 4.2.1. N-PLS Method on the Original EEM Spectra

As for Data Set 1, the optimal *LV*s determined for Val–Tyr–Val, Trp–Gly, and Phe were three, two, and five, respectively, and the corresponding statistical parameters are illustrated in Table 1. As for Data Set 2, the optimal *LV*s determined for magnolol and honokiol were four and five, respectively, and the corresponding statistical parameters are illustrated in Table 2.

As illustrated in Table 1 and Table 2, the established models have high values of *R_c_* and *R_loo-cv_* (higher than 0.9693), and low values of *RMSEC* and *RMSECV* (lower than 5.5521), showing that these models have good linearity, high reliability, and accuracy. Moreover, the values of *R_p_* and *RMSEP* are higher than 0.9921 and lower than 3.8600, respectively, indicating that the established models have a good predictive ability.

#### 4.2.2. PLS Method on the Extracted Maximum Spectra in the Concatenated Mode

As for Data Set 1, the optimal *LV*s determined for Val–Tyr–Val, Trp–Gly, and Phe were three, four, and five, respectively, and the corresponding statistical parameters are illustrated in Table 1. As for Data Set 2, the optimal *LV*s determined for magnolol and honokiol were five and four, respectively, and the corresponding statistical parameters are illustrated in Table 2.

As illustrated in Table 1 and Table 2, the established models have high values of *R_c_* and *R_loo-cv_* (higher than 0.9853), and low values of *RMSEC* and *RMSECV* (lower than 2.8213), showing that these models have good linearity, high reliability, and accuracy. Moreover, the values of *R_p_* and *RMSEP* are higher than 0.9684 and lower than 10.1635, respectively, indicating that the prediction ability of the model is not poor. As can be seen from Table 1 and Table 2, the results of the N-PLS method using the new constructed 3D EEM spectra are comparable to those of the N-PLS method on the original EEM spectra and the PLS method using the extracted maximum spectra in the concatenated mode. These obtained results demonstrate that the characteristic spectra were effectively extracted from the original EEM spectra.

To further check the accuracy of the N-PLS models on the new constructed 3D EEM spectra, the N-PLS models on the original EEM spectra, and the PLS models on the extracted maximum spectra in the concatenated mode, an analytical recovery experiment for Data Set 2 was carried out using a standard addition method, and the results are summarized in Table 3. The concentrations predicted using the N-PLS models using the new constructed 3D EEM spectra showed a recovery of 95.6% for magnolol and 85.6% for honokiol. The N-PLS models using the original EEM spectra showed a recovery of 91.0% for magnolol and 86.7% for honokiol. The PLS models on the extracted maximum spectra in the concatenated mode showed a recovery of 89.6% for magnolol and 88.1% for honokiol. The obtained results demonstrated that the proposed method has better recovery and precision for magnolol and honokiol. The above results demonstrated that the proposed strategy showed acceptable analytical performance in the real analytical utility.

## 5. Conclusions

In the present contribution, the N-PLS method on the new constructed 3D EEM spectra was proposed and applied in two Data Sets. In order to display real spectral information and guarantee the model, the resulting spectral data were preprocessed, such as via scatter removal. The correlation coefficients of prediction based on N-PLS on the new constructed 3D EEM was above 0.9906, and the recovery values were in the range of 85.6–95.6%. Therefore, one can conclude that the N-PLS method combined with the new constructed 3D EEM spectra is expected to be broadened as an alternative strategy for the simultaneous determination of multiple target compounds. Moreover, the comparison of results with the N-PLS method on the original EEM spectra and the PLS method on the extracted maximum spectra in the concatenated mode demonstrated that the satisfactory results of the proposed strategy are attributed to the proper extraction of the characteristic spectra. In conclusion, this study has shown that the proposed approach can open a new window to the analysis of the EEM spectra in the presence of signal overlap.

## Figures and Tables

**Figure 1 molecules-27-08378-f001:**
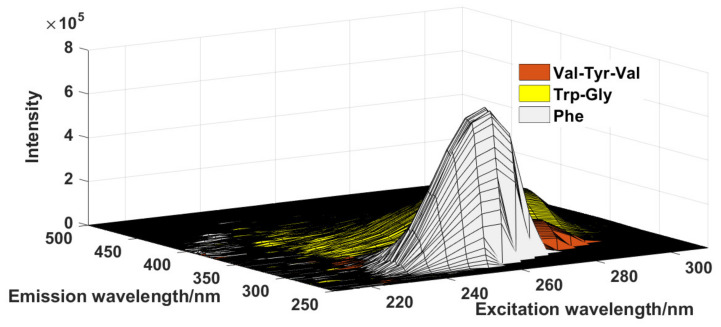
The EEM spectra for each target compound in Data Set 1.

**Figure 2 molecules-27-08378-f002:**
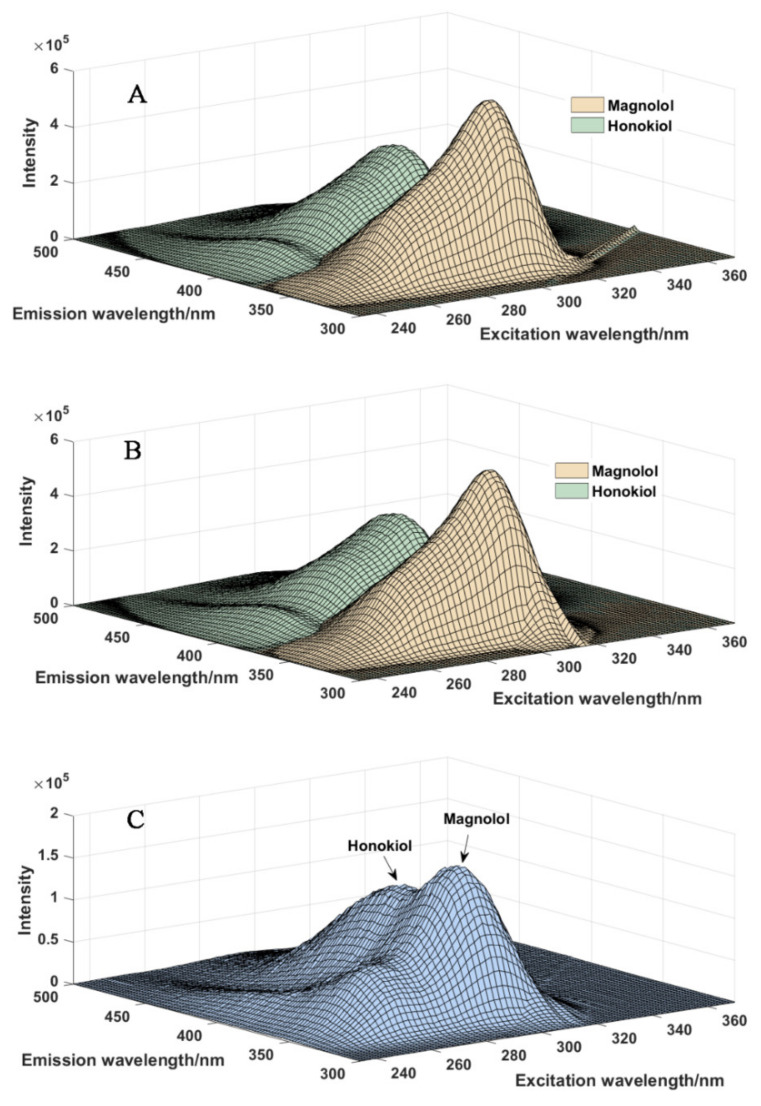
The EEM spectra for pure magnolol and honokiol with scatterings (**A**), without scatterings (**B**), and Magnoliae Cortex (**C**).

**Figure 3 molecules-27-08378-f003:**
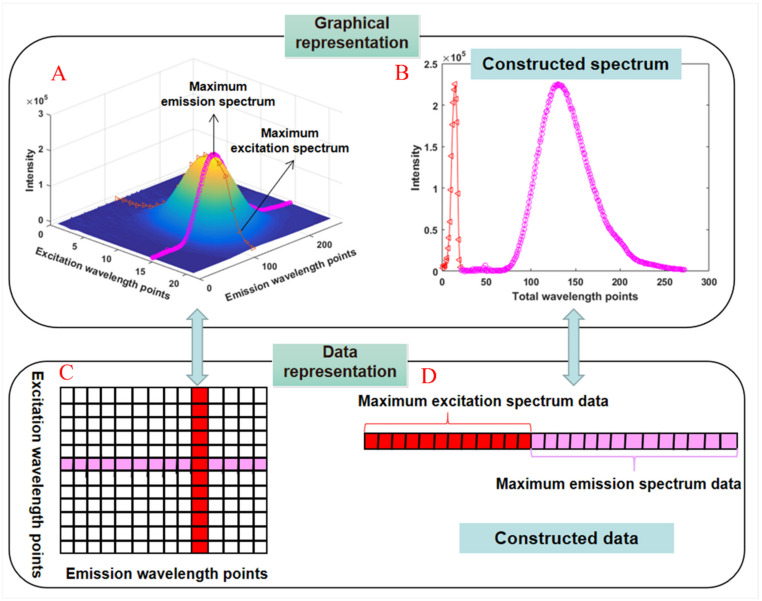
The schematic diagram for the constructed spectrum, the maximum excitation and emission spectra for each target compound (**A**), the extracted maximum excitation spectrum and maximum emission spectrum (**B**), the data of each extracted spectrum (**C**), the representation of the data matrix (**D**).

**Figure 4 molecules-27-08378-f004:**
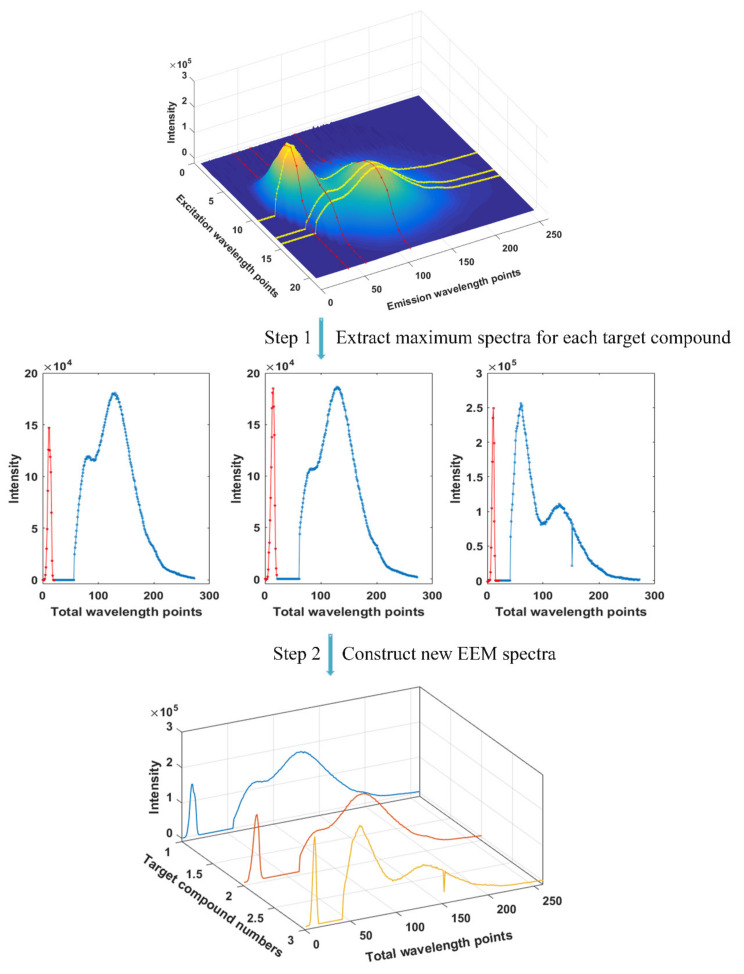
The overall strategy for the construction of the new constructed 3D EEM spectra.

**Figure 5 molecules-27-08378-f005:**
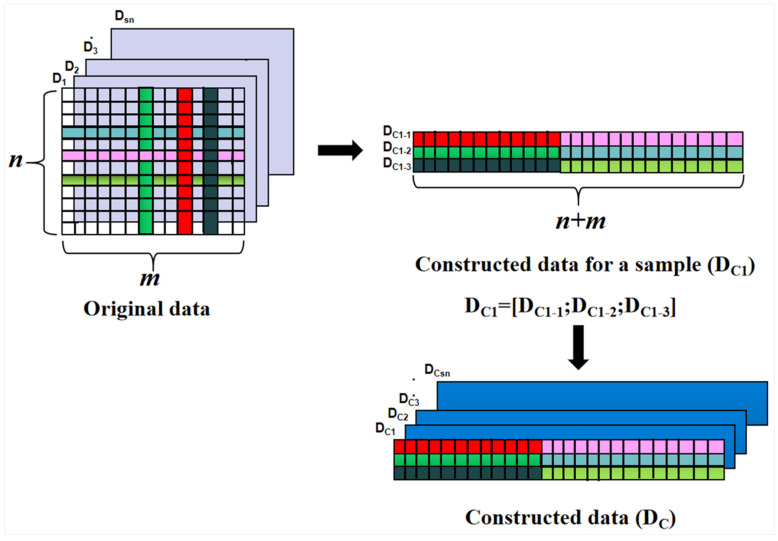
The process of the data construction.

**Figure 6 molecules-27-08378-f006:**
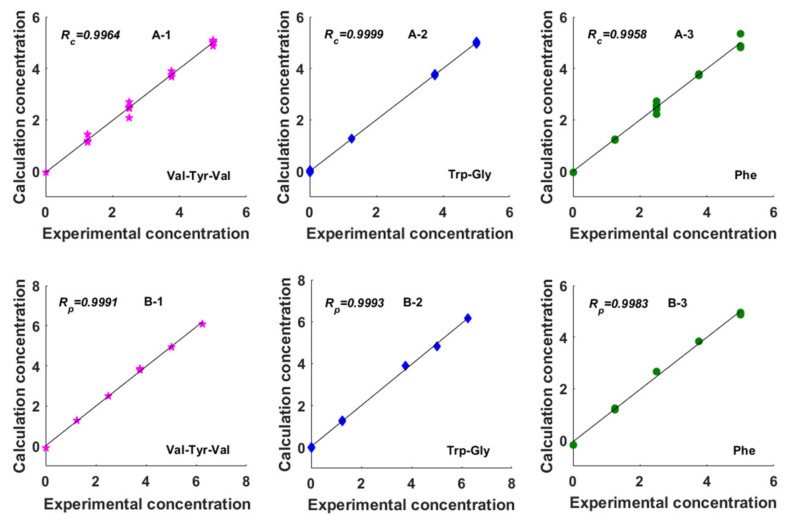
Plot of the experimental concentration against the calculation concentration obtained from the N-PLS method on the basis of the new constructed 3D EEM spectra for Data Set 1. A-1, A-2, and A-3 are the correlation coefficient of calibration (*R_c_*) for Val–Tyr–Val, Trp–Gly, and Phe, respectively. B-1, B-2, and B-3 are the correlation coefficient of prediction (*R_p_*) for Val–Tyr–Val, Trp–Gly, and Phe, respectively.

**Figure 7 molecules-27-08378-f007:**
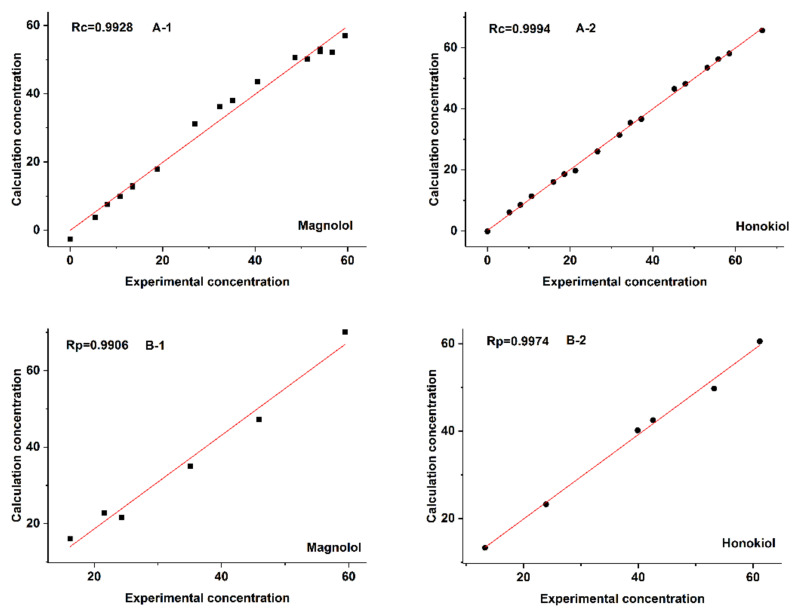
Plot of the experimental concentration against the calculation concentration obtained from the N-PLS method on the basis of the new constructed 3D EEM spectra for Data Set 2. A-1 and A-2 are the correlation coefficient of calibration (*R_c_*) for magnolol and honokiol, respectively. B-1 and B-2 are the correlation coefficient of prediction (*R_p_*) for magnolol and honokiol, respectively.

**Table 1 molecules-27-08378-t001:** The statistical parameters of the N-PLS models using the new constructed 3D EEM spectra, the N-PLS models using the original EEM spectra, and the PLS models using the extracted maximum spectra in the concatenated mode of Data Set 1.

Method	Item	Val–Tyr–Val	Trp–Gly	Phe
*N-PLS*^1^	*LV*s	3	5	5
	*R_c_*	0.9964	0.9999	0.9958
	*RMSEC*	0.1333	0.0254	0.1339
	*R_loo-cv_*	0.9951	0.9997	0.9872
	*RMSECV*	0.1553	0.0487	0.2335
	*R_p_*	0.9991	0.9993	0.9983
	*RMSEP*	0.0901	0.0961	0.1111
*N-PLS*^2^	*LV*s	3	2	5
	*R_c_*	0.9957	0.9967	0.9925
	*RMSEC*	0.1457	0.1756	0.1778
	*R_loo-cv_*	0.9941	0.9957	0.9808
	*RMSECV*	0.1453	0.1706	0.2512
	*R_p_*	0.9999	0.9869	0.9987
	*RMSEP*	0.0812	0.3944	0.0940
PLS	*LV*s	3	4	5
	*R_c_*	0.9969	0.9999	0.9963
	*RMSEC*	0.1208	0.0278	0.1241
	*R_loo-cv_*	0.9948	0.9997	0.9853
	*RMSECV*	0.1562	0.0511	0.2527
	*R_p_*	0.9992	0.9993	0.9984
	*RMSEP*	0.0881	0.0972	0.1020

*N-PLS*^1^: N-PLS models using the new constructed 3D EEM spectra. *N-PLS*^2^: N-PLS models using the original EEM spectra.

**Table 2 molecules-27-08378-t002:** The statistical parameters of the N-PLS models using the new constructed 3D EEM spectra, the N-PLS models using the original EEM spectra, and the PLS models using the extracted maximum spectra in the concatenated mode of Data Set 2.

Method	Item	Magnolol	Honokiol
*N-PLS*^1^	*LV*s	4	4
	*R_c_*	0.9928	0.9994
	*RMSEC*	2.3957	0.7081
	*R_loo-cv_*	0.9780	0.9976
	*RMSECV*	4.7487	1.4362
	*R_p_*	0.9906	0.9974
	*RMSEP*	4.5444	1.4700
*N-PLS*^2^	*LV*s	4	5
	*R_c_*	0.9894	0.9993
	*RMSEC*	2.8895	0.7442
	*R_loo-cv_*	0.9693	0.9962
	*RMSECV*	5.5521	1.8324
	*R_p_*	0.9921	0.9978
	*RMSEP*	3.8600	1.4685
PLS	*LV*s	5	4
	*R_c_*	0.9974	0.9974
	*RMSEC*	1.4284	1.4275
	*R_loo-cv_*	0.9905	0.9919
	*RMSECV*	2.8213	2.6537
	*R_p_*	0.9684	0.9781
	*RMSEP*	10.1635	4.0459

*N-PLS*^1^: N-PLS models using the new constructed 3D EEM spectra. *N-PLS*^2^: N-PLS models using the original EEM spectra.

**Table 3 molecules-27-08378-t003:** Determination results of magnolol and honokiol in Magnoliae Cortex sample using different methods.

Compound	Method	Magnoliae Cortex (μg/mL)	Added Sample (μg/mL)		
			Added	Predicted	Recovery (%)
Magnolol	*N-PLS*^1^	9.0		34.8	95.6
	*N-PLS*^2^	10.1	27.0	34.7	91.0
	PLS	11.7		35.9	89.6
Honokiol	*N-PLS*^1^	6.6		29.3	85.6
	*N-PLS*^2^	6.8	26.6	29.9	86.7
	PLS	8.9		32.3	88.1

*N-PLS*^1^: N-PLS models using the new constructed 3D EEM spectra. *N-PLS*^2^: N-PLS models using the original EEM spectra.

## Data Availability

Not applicable.

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
