# Peer review of "New Constructed EEM Spectra Combined with N-PLS Analysis Approach as an Effective Way to Determine Multiple Target Compounds in Complex Samples"

_molecules, 2022, doi:10.3390/molecules27238378_

Round 1

Reviewer 1 Report

Line 3: the term “absorption” would be more correct if exchanged for “extinction.”

Description of Data sets in Section 2: the authors do not state if the raw data were (1) corrected for inner filter effects (IFEs), (2) normalized to Raman or quinine units, or (3) if the scatter below an excitation wavelength of 250 nm was omitted in the calculations (see Fig. 1). These preprocessing steps, applied to EEM spectra before further statistical analyses are performed, are commonly accepted practices in the EEM community in the literature. If these steps were not conducted, the manuscript is not suitable for publication.

In Fig. 1, the region in the spectrum in which Excitation wavelengths exceed Emission wavelengths should not be depicted in the same color as the Phe analyte.

Author Response

Response to the reviewer’s comments:

Reviewer 1

Comments and Suggestions for Authors

Line 3: the term “absorption” would be more correct if exchanged for “extinction.”

This mistake has been corrected.

Description of Data sets in Section 2: the authors do not state if the raw data were (1) corrected for inner filter effects (IFEs), (2) normalized to Raman or quinine units, or (3) if the scatter below an excitation wavelength of 250 nm was omitted in the calculations (see Fig. 1). These preprocessing steps, applied to EEM spectra before further statistical analyses are performed, are commonly accepted practices in the EEM community in the literature. If these steps were not conducted, the manuscript is not suitable for publication.

Thanks for your suggestion! In data set 1, as described in the download website, in order to avoid 1st order Rayleigh scatter, emission data were not acquired in an interval of 10 nm from the excitation wavelength, therefore, the downloaded spectral data without scatterings have been used. In data set 2, we use interpolation method to remove the scattering. By the application of the interpolation method, the width of the scatter areas must be assessed. This was done by visual inspection of the data and confirmed subsequently by plotting parts of the preprocessed data. Very large widths will cause some uncertainty in the interpolated area whereas too narrow widths will bias the solution because scatter will be included. Approximately 1.5 times of the visually assessed scatter area was removed in order to completely remove the scatter values. We recalculated the data after removing scattering, and the results are shown in the manuscript.

In Fig. 1, the region in the spectrum in which Excitation wavelengths exceed Emission wavelengths should not be depicted in the same color as the Phe analyte.

We have redrawn Fig. 1.

Reviewer 2 Report

The manuscript is concerned with the N-PLS method on the new constructed 3D EEM spectra. The method was proposed to quantify target compounds in two complex data sets, and good quantitative results were obtained. However, there are some questions which should be addressed, as listing below:

1. From the tabular data, there is no feeling that the new method is superior to the other two methods (N-PLS models 355 using original EEM spectra and PLS models).

2. The N-PLS does not have second-order advantages, how to eliminate unknown interferences. At present, the results of the data set 1 and 2 are good, is it because the background interference is small? If in the plasma system, the method proposed by the authors can also accurately quantify the target concentration.

3. It is suggested to compare N-PLS with ATLD or PARAFAC. These are classic EEM data matrix processing methods.

4. Figure 1 is wrong.

5. It is suggested that scattering should be deducted before EEM data matrix processing to improve the accuracy of results

6. There are spelling errors in the article, such as "mathods" in Table 3. Please inspect the article carefully.

Author Response

Reviewer 2

Comments and Suggestions for Authors

The manuscript is concerned with the N-PLS method on the new constructed 3D EEM spectra. The method was proposed to quantify target compounds in two complex data sets, and good quantitative results were obtained. However, there are some questions which should be addressed, as listing below:

Thanks for your encouragement! We have revised our manuscript according to your comments and the answers to your questions have been listed as follows one by one!

  1. From the tabular data, there is no feeling that the new method is superior to the other two methods (N-PLS models 355 using original EEM spectra and PLS models).

In our calculation results, we can find that the new method can get the same or better results as the original data. This method does not lose the important information of the compounds at the excitation wavelength and emission wavelength in the EEM spectrum, and can open a new window for the EEM spectrum analysis with signal overlap.

  1. The N-PLS does not have second-order advantages, how to eliminate unknown interferences. At present, the results of the data set 1 and 2 are good, is it because the background interference is small? If in the plasma system, the method proposed by the authors can also accurately quantify the target concentration.

      Thank you for your suggestion. In the present study, the N-PLS method can obtain relatively satisfactory results, we think there should be two reasons. On the one hand, we have adopted a new method to construct a three-dimensional spectral, which makes the newly constructed three-dimensional spectra contain less unknown interference information. On the other hand, there may be less interference information in the used data sets. We are going to apply this algorithm to the quantitative analysis of target substances in the plasma system in the next step.

  1. It is suggested to compare N-PLS with ATLD or PARAFAC. These are classic EEM data matrix processing methods.

       Thank you for your suggestion. The ATLD and PARAFAC methods are classic EEM data matrix processing methods. However, it is a pity that we have not learned to use these methods at present. In the future study, we will make up for these shortcoming.

  1. Figure 1 is wrong.

 We have redrawn Fig. 1.

  1. It is suggested that scattering should be deducted before EEM data matrix processing to improve the accuracy of results

Thanks for your suggestion! In the revised manuscript, we use interpolation method to remove the scattering, and the quantitative analysis has been performed on the EEM data matrix that without scatterings. The newly obtained results have been supplemented in the revised manuscript.

  1. There are spelling errors in the article, such as "mathods" in Table 3. Please inspect the article carefully.

We rechecked the article and the spelling mistakes in the article have been corrected.

Round 2

Reviewer 1 Report

In my previous review of this paper, I reported that “the authors do not state if the raw data were (1) corrected for inner filter effects (IFEs), (2) normalized to Raman or quinine units, or (3) if the scatter below an excitation wavelength of 250 nm was omitted in the calculations (see Fig. 1). These preprocessing steps, applied to EEM spectra before further statistical analyses are performed, are commonly accepted practices in the EEM community in the literature.

In the authors response, they appropriately addressed the removal of scatter in their research, but they did not address whether they corrected for IFEs or normalized their spectra to Raman or quinine units. Because in their response to my review they did not address these points, I can only assume that they did not conduct these procedures in their research. Therefore, I must reject this manuscript for publication.

as for the 2nd round comments, The authors replied, you may find it in the attached file

Reviewer 2 Report

It's all OK.